# A Green and Facile Microvia Filling Method via Printing and Sintering of Cu-Ag Core-Shell Nano-Microparticles

**DOI:** 10.3390/nano12071063

**Published:** 2022-03-24

**Authors:** Guannan Yang, Shaogen Luo, Tao Lai, Haiqi Lai, Bo Luo, Zebo Li, Yu Zhang, Chengqiang Cui

**Affiliations:** 1State Key Laboratory of Precision Electronic Manufacturing Technology and Equipment, Guangdong University of Technology, Guangzhou 510006, China; ygn@gdut.edu.cn (G.Y.); 2111901339@mail2.gdut.edu.cn (S.L.); 1111901010@mail2.gdut.edu.cn (T.L.); 2111901120@mail2.gdut.edu.cn (H.L.); 2112101330@mail2.gdut.edu.cn (B.L.); 3119000337@mail2.gdut.edu.cn (Z.L.); 2Jihua Laboratory, Foshan 528225, China

**Keywords:** Cu-Ag core-shell nano-microparticles, microvias, vertical interconnection, blind hole filling, sintering

## Abstract

In this work, we developed an eco-friendly and facile microvia filling method by using printing and sintering of Cu-Ag core-shell nano-microparticles (Cu@Ag NMPs). Through a chemical reduction reaction in a modified silver ammonia solution with L-His complexing agent, Cu@Ag NMPs with compact and uniform Ag shells, excellent sphericity and oxidation resistance were synthesized. The as-synthesized Cu@Ag NMPs show superior microvia filling properties to Cu nanoparticles (NPs), Ag NPs, and Cu NMPs. By developing a dense refill method, the porosity of the sintered particles within the microvias was significantly reduced from ~30% to ~10%, and the electrical conductivity is increased about twenty-fold. Combing the Cu@Ag NMPs and the dense refill method, the microvias could obtain resistivities as low as 7.0 and 6.3 μΩ·cm under the sintering temperatures of 220 °C and 260 °C, respectively. The material and method in this study possess great potentials in advanced electronic applications.

## 1. Introduction

Microvia filling is an essential technology for the interconnection metallization of integrated circuits (ICs), printed circuit boards (PCBs), and micro-electromechanical systems (MEMS), as well as 3D electronic packaging [1,2,3]. The microvia filling technology enables the vertical electrical interconnections, and thus, improving the communication rate and reducing the volume and power consumption of electronic devices.

To date, electroplating is still the predominant method for microvia filling. The electroplated microvias could obtain satisfactory electrical properties, however, this technique has several drawbacks: (1) The electroplating is a multi-step process composed by exposing, developing, etching, and plating procedures, which are complex and the by-products may cause environment pollution [4]. (2) For electroplated microvias with large depth/diameter ratio, it is difficult to achieve the “bottom-up” filling mode and voids will form inside the microvias [5]. As the current density is limited to maintaining stable plating, the efficiency of electroplated microvias with large diameter and depth is low [6]. With the rapid development of electronic industry, eco-friendly, fast and flexible microvia filling techniques are urgently needed and expected to be the substitute for traditional electroplating techniques.

Selective printing and sintering of metallic NMPs might provide a possible approach for this issue [7,8]. Metallic (Cu, Ag, etc.,) particles are printed on a substrate and then sintered by a laser source or heat source. Via selective printing or laser sintering, the specifically designed patterns could be obtained. Due to the extremely high specific area and surface energy, the required sintering temperature of metal nanoparticles is only 200–300 °C [9], which enables the printing of electronic circuits on heat-sensitive substrates. In comparison with traditional electroplating technique, the selectively printing/sintering of metallic particles significantly reduces the processing period and the amount of toxic byproducts, and therefore, it is more efficient and eco-friendly. In recent years, the method has been extensively used in the investigations of printed electronic circuits with various conductive materials, line widths/spacings, upon diverse substrates, and in specific application scenarios [10,11,12,13,14,15]. The electrical conductivity of the as-sintered metallic particles could reach the order of 10 μΩ·cm [16].

Nevertheless, the studies on microvia filling via selective printing and sintering of metallic particles are still far from enough [3,17]. Cho et al. attempted to fill blind vias with 100 μm in diameter by silver nanoparticle ink [18], Khorramdel et al. tried to fill microvias in silicon substrates by silver nanoparticle ink [3]. However, the resistivity of the filled microvias is 2–4 orders of magnitude higher than that of bulk copper, which is undesirable in electronic applications. Due to the porous structure of the sintered metallic particles, the conductivity of as-sintered metal nanoparticles is lower than that of the bulk metals. For the case of microvia filling, it is difficult to compactly fill metallic particles into the microvias and apply pressure to the metallic particles during sintering. Due to the solidification shrinkage of the particles, considerable pores will form inside the microvias, further deteriorating the electrical/mechanical properties and reliability [19]. Finding a path to improve the compactness and sintering performance of microvia filling with metallic particles is critical to their widespread application.

In this study, we try to improve the microvia filling performance of metallic particles from the perspective of material design and filling method. In terms of filling materials, Cu and Ag particles are widely used in the previous studies of conductive inks, interconnect materials, and printed electronics [15,20,21]. However, Cu particles are easily oxidized in air, and Ag particles are prone to electromigration, which severely limit their applications [22]. Here we propose a chemical reduction method to synthesize Cu@Ag NMPs as microvia fillers. The as-synthesized Cu@Ag NMPs can achieve the combined properties of oxidation and electromigration resistance. The Cu@Ag NMPs are in a wide size range from 0.2~1.4 μm, which can reach higher packing density than uniform-sized particles. To further improve the filling compactness, the Cu@Ag NMPs are mixed with volatile, low-viscosity, and decomposable ethylene glycol solvent to prepare a paste with a reactively high solid content (80 wt.%). Compared with other metal particle conductive inks with relatively low solid content (40~50 wt.% [23]), and metal particle conductive adhesives with undecomposable organic additives such as film-forming agents and cross-linking agents [24], the Cu@Ag particle paste is able to reach better sinterability and conductivity. In terms of filling method, previous studies have tried screen printing [25,26,27], squeegee printing [28], and inkjet printing [7]. However, the filling performance of these methods are unsatisfactory, as the evaporation of solvents will lead to shrinkage cavities in the structure. Here we develop a dense filling method for microvias. Through drying and refilling the microvias, a better filling compactness can be obtained. With the combined improvements in material and filling method, a promising microvia filling performance can be achieved.

## 2. Experiments

### 2.1. Materials Preparation

Cu NMPs (Aladdin Reagent Co., Ltd., Shanghai, China) with the size of (0.8 ± 0.6) μm and Ag NPs (analytical grade, 99.5%, Shanghai Macklin Biochemical Technology Co., Ltd., Shanghai, China) with the diameter of (60–120) nm were employed. The chemicals used in this work are listed below: (1) ethylene glycol ((CH_2_OH)_2_, analytical grade, 99%, Sinopharm Chemical Reagent, Shanghai, China). (2) Silver nitrate (AgNO_3_, analytical grade, 99%, HUADA Reagent). (3) Ammonia (NH_3_·H_2_O, analytical grade, 25%, DAMAO Chemical Reagent). (4) Absolute ethanol (analytical grade, 99.7%, Sinopharm Chemical Reagent). (5) L-His (C_6_H_9_N_3_O_2_, analytical grade, 99%, Aladdin Biochemical Technology Co., Ltd., Shanghai, China). (6) polyethylene glycol (PEG, average molecular weight 1000, HO(CH_2_CH_2_O)_n_H, analytical grade, 99%, Aladdin Biochemical Technology Co., Ltd., Shanghai, China). All chemicals were used as received without further purification.

First, 0.2 mol/L Cu NMPs, 0.2 mol/L L-His, 0.02 mol/L PEG, and 0.6 mol/L ammonia were adequately mixed and dissolved in 50 mL aqueous solution. Second, 0.02 mol/L AgNO_3_, 0.2 mol/L L-His, 0.02 mol/L PEG, and 0.8 mol/L ammonia were dissolved in another 200 mL aqueous solution. Third, the solution with Cu NMPs was added into the solution with AgNO_3_ and ultrasonically vibrated and mechanically stirred for 30 min. Then, Cu@Ag NMPs were extracted from the solution through centrifugation at 12,000 rpm for 5 min and cleaned by deionized water for three times and ethanol for one time. After vacuum drying, the Cu@Ag NMPs were finally obtained. Cu@Ag NMP pastes were prepared by mixing the as-synthesized Cu@Ag NMPs with 20 wt.% ethylene glycol.

### 2.2. Filling Method

Flexible copper clad laminate (FCCL) with blind vias was used as the substrate for microvia filling. The thickness of the FCCL is 70 μm, including 30 μm-thick polyimide (PI) in the center and 20 μm-thick Cu laminate on both surfaces, as shown in Figure 1a. The depth and diameter of the blind vias are 50 μm and 75 μm, respectively. Using a P3000H machine at the power of 7000 W, the substrate was cleaned by CF_4_/O_2_ plasma for 10 min to remove the impurities. The Cu@Ag NMP pastes were filled into the microvias through a dense refill process as shown in Figure 1. Here, we describe all steps of the process in detail: (1) the Cu@Ag NMP pastes were printed on the top of the microvias by a nozzle (Figure 1b). (2) The samples were kept in vacuum chamber for 3 min, during which the air inside of microvias will escape and the pastes will permeate into the microvias due to the capillary phenomenon (Figure 1c). (3) Upper surface was roll pressed to further squeeze the paste into the microvias (Figure 1d). (4) The Cu@Ag NMP paste was re-printed on top of the microvias (Figure 1e) and the sample was dried in 80 °C air gas flow for 3 min (Figure 1f), and a part of the ethylene glycol solution in the paste evaporated. (5) The samples were roll pressed again to further squeeze the paste into the microvias (Figure 1g). Finally, the samples were sintered in 5%H_2_ + 95%Ar atmosphere by a hot-pressing furnace (Figure 1h). The pressure was 2 MPa and the sintering temperatures were 220 °C. Figure 2 shows a typical temperature curve during the sintering process.

### 2.3. Characterization

A scanning electron microscope (SEM, HITACHI SU8220) and a high-resolution transmission electron microscope (HR-TEM, FIB Talos F200S) were used to observe the morphologies of the as-synthesized Cu@Ag NMPs and the filled microvias after sintering. The element distribution of the as-synthesized Cu@Ag NMPs was detected by an energy dispersive spectrometer (EDS, BRUKER XFlash 6|30). The size distribution of the as-synthesized Cu@Ag NMPs was measured by a laser particle size analyzer (Mastersizer 3000, Malvern, UK). The phase composition of the as-synthesized Cu@Ag NMPs was characterized by an X-ray diffractometer (XRD, Bruker D8 ADVANCE, Karlsruhe, Germany) with a scan rate of 10°/min in the 2*θ* range of 20–80°. The thermogravimetry (TG) curves of the particles were measured in air by a TGA/DSC3+ system (Mettler Toledo, Zurich, Switzerland) at the heating rate of 10 °C/min. Using a test platform (Guangzhou Four Probe Technology Co., Ltd., ST-102E, Guangzhou, China) with a source meter unit (Keithley, 6220) and a nanovoltmeter (Keithley, 2182A), the resistivity of the sintered microvias was measured by four-point probe method [29]. Figure 3 shows a schematic of the four-point probe measurement method.

## 3. Results and Discussions

### 3.1. Micro-Structures of Cu@Ag NMPs

Figure 4 shows the SEM images of the Cu NMPs and as-synthesized Cu@Ag NMPs. The particles have smooth surface and are uniformly dispersed without obvious aggregation. The size distributions of the two kinds of particles are basically consistent. The particle size is in the range of 0.2~1.4 μm and the average size is 0.8 μm. Figure 5 shows the TEM images of the as-synthesized Cu@Ag NMPs. The enlarged image of an individual particle shows that the particle is in quasi-spherical shape (Figure 5b). Figure 5c–e shows the EDS maps of the Cu@Ag particle. It can be found that the Ag element distributes uniformly on the surface areas. The Cu element, however, distributes inside the particle. Figure 5f shows the Cu and Ag element distribution curves across the diameter of the Cu@Ag particle. It can be seen that the signal of Ag symmetrically forms two peaks at the edges, while the signal of Cu forms a peak nearby the central area. These results confirm that the as-synthesized Cu@Ag NMPs have a uniform Cu core-Ag shell structure. The thickness of the Ag shell is about 50–100 nm.

These results confirm the promising coating quality of the as-synthesized Cu@Ag NPs. Before this study, extensive efforts have been made to prepare Cu@Ag NPs [30,31,32,33]. However, due to the large potential difference in the Cu-Ag replacement reaction (Δ*E* = +0.4621 V [34]), it is difficult to obtain a densely coated silver layer under normal conditions. Many previous synthetic methods are carried out in organic solvents, and required inert/reducing gas protection, high reaction temperature (Tens to hundreds of degrees Celsius), and long reaction time (several hours) [35,36,37]. In comparison, the method of this study is much more convenient and eco-friendly. In our previous study, we pointed out that the addition of complexing agent could make the deposition potential of metal ions shift negatively and thereby improve the coating quality of core-shell bimetallic particles [38]. The L-His additive used in this study has been proven as an effective complexing agent for metal ions, and has been applied in the synthesis of porous metal–organic frameworks [39]. With the usage of L-His, the synthesis process of Cu@Ag NMPs is facilitated. It is a one-step reduction reaction within only several minutes, using aqueous solution instead of organic solution, without heating and atmosphere protection.

### 3.2. Oxidation Resistance of Cu@Ag NMPs

Figure 6a shows the XRD patterns of the as-synthesized Cu@Ag NMPs before and after 3 months’ exposure in the atmosphere. For comparison, we obtained the XRD patterns of the ~300 nm diameter Cu NMPs and Cu NPs before and after 15 days’ exposure in the atmosphere. It can be found that, after 3 months’ exposure in the atmosphere, the XRD pattern of Cu@Ag NMPs shows the peaks of crystalline Cu and Ag without any peaks of oxides. For the uncoated Cu NMPs, due to their relatively poor oxidation resistance, oxide peaks of Cu_2_O can be detected after 3 months’ air exposure. The Cu nanoparticles are more severely oxidized and show stronger oxidation peaks. The results indicate that smaller Cu particles are more easily oxidized than the larger ones. Silver layer coating significantly improves the oxidation resistance of the Cu NMPs during storage and at relatively low temperature, which reflects the excellent quality of the Ag shells and advantage of this method.

Figure 6b compares the TG curves of the as-synthesized Cu@Ag NMPs, Cu NMPs, and Cu nanoparticles. The particles start to gain weight at the temperature of 150–200 °C as the oxidation reaction started, and the gained weight will be a constant as the oxidation reaction finished. In comparison with the Cu nanoparticles, the weight gain of the Cu@Ag NMPs and Cu NMPs is much slower. It confirms that the use of larger particle is an effective way to improve the oxidation resistance.

### 3.3. Characterization of Microvia Filling Performance

Figure 7 shows the SEM images of the microvias filled with Cu@Ag NMPs at the sintering temperature of 220 °C and sintering time of 10 min, 30 min, and 50 min. Figure 8 shows the measured electrical resistivities of the microvias. It can be observed that the microvias are fully filled with the particles. For the sample sintered for 10 min, the electrical resistivity of the sample is measured as 51 μΩ·cm. The SEM images show that there are still gaps between the original Cu laminates and particle paste. A large number of spherical particles can be observed, indicating that the particles are insufficiently sintered and have not merged with the Cu laminates at this temperature. For the sample sintered for 30 min and 50 min, the electrical resistivities of the samples decrease to 8.2 μΩ·cm and 7.0 μΩ·cm, respectively. The SEM images show that the particles have merged with each other and the original Cu laminates. It is consistent with the relatively good sintering performance at these temperatures.

### 3.4. Advantage of Cu@Ag NMPs in Microvia Filling

In previous reports, Khorramdel et al. [3] tried to fill blind vias on silicon substrates with Ag NPs paste under the sintering temperature of 220 °C. But the resistivity of the filled vias are 2–4 orders of magnitude higher than bulk silver. Yang et al. [40] filled TSVs with silver NPs paste under the sintering temperature of 220 °C, and the resistivity of the filled vias is relatively high (400 μΩ·cm). With increasing sintering temperature, the sintering properties of metal particles can be significantly improved [41]. Quack et al. [42] filled TSVs with Ag NPs under the sintering temperature of 250 °C, and the resistivity is less than 200 μΩ·cm. In our previous work [41], we found that sintering strength of Cu MNPs increased by 100~230% when the sintering temperature increased from 220 °C to 260 °C. Under the sintering temperature of 220 °C, the MNPs are unable to form thick sintering necks with each other. Therefore, in this work, we chose the sintering temperature of 260 °C to better compare the filling performance of different materials and different filling methods.

Figure 9 compares the cross-section images of the microvias filled with Cu NPs, Ag NPs, Cu NMPs, and Cu@Ag NMPs at the sintering temperature of 260 °C and sintering time of 30 min. It can be found that all the samples have been densely filled with the particles, and there are no obvious gaps between the Cu laminate and the particle paste. The electrical resistivities of the samples were measured as 21.3, 13.9, 10.6, and 6.3 μΩ·cm respectively, as shown in Figure 10.

In comparison with Cu NPs, the Ag NPs exhibited lower electrical resistivity, indicating the optimal sintering property of Ag. It might be caused by the low melting point and excellent oxidation resistance of Ag. Previous research has also documented that the sintering property of Ag NPs is superior to that of Cu NPs [37,43,44]. In comparison with Cu NPs, the Cu NMPs exhibited lower electrical resistivity owing to their wide size distribution. For the particle paste with mixed small and large particles, the small particles could fill the gaps between the large particles and hence the sintered structure owns higher compactness and better properties. It has been reported that the particle paste with mixed particles size could achieve better properties than the particle paste with uniform particle size [45,46]. The Cu@Ag NMPs combine the advantages of Ag shell and mixed particle sizes, thus exhibiting better conductivity among the four groups of particles.

### 3.5. Effect of Dense Refill Method

Before this study, a few research attempted to fill microvias using metallic particles. However, the electrical property of the filled microvias is unsatisfactory. Cho [18] and Khorramdel [3] obtained microvias with electrical resistances of 0.1 Ω and 4 Ω. The corresponding electrical resistivities of the microvias were calculated as 785 μΩ·cm and 1.94 × 10^4^ μΩ·cm respectively, which are 2–4 orders of magnitude higher than that of bulk copper. In these studies, the particle pastes are filled into the microvias using the traditional stencil printing method. Considerable large-scale voids or gaps could be observed in the structure after sintering, which deteriorate the compactness and property of the microvias.

In this study, we proposed a dense refill method to improve the compactness of the microvia filled with metallic particles. Figure 11 compares the cross-section images of the microvias filled by the stencil printing and the dense refill method using Cu@Ag NMPs at a sintering temperature of 260 °C. For the stencil printing method, there are numerous pores of tens of microns in size inside the microvia, and an obvious gap between the sintered structure and the original Cu laminate. The porosity area fraction of cross-section is ~20%. For the dense refill method, the size of the pores is significantly reduced to ~1 μm, and the gap between the sintered structure and the original Cu laminate is reduced. The porosity area fraction is reduced to ~8%. The resistivity of the microvias was also significantly reduced by ~95% from 131.6 μΩ·cm to 6.3 μΩ·cm. 

The advantages of the dense refill method are discussed as below: during the wet filling process (step b–d in Figure 1), the wet paste is pressed into the microvia. The capillary phenomenon results in complete filling of the wet paste into the microvia. By contrast, dry particles can hardly completely fill the microvia, as the particles adhere to the sidewall of the microvias owing to the high surface tension and internal friction of the particles. Although wet paste fills the microvia better, the space occupied by the solution in the paste increases the gaps between the particles, and then forms pores after sintering. Therefore, after the wet filling process, the samples are re-dried (Figure 1f), during which a part of the ethylene glycol solution in the paste evaporates. Then the samples are roll pressed to squeeze the paste into the microvias and compensate the voids (Figure 1g). By this dry-refilling process, the compactness and sintering performance of the microvia can be significantly improved.

## 4. Conclusions

In summary, we proposed an eco-friendly and facile microvia filling method through printing and sintering of Cu@Ag NMPs. Cu@Ag NMPs with compact and uniform Ag shells, good sphericity and promising oxidation resistance were obtained. The as-synthesized Cu@Ag NMPs show optimal microvia filling property. Using a dense refill method, the compactness of the particles in the microvias is significantly improved and the resistivity is reduced by approximately ~95%. With the combination of the Cu@Ag NMPs and the dense refill method, the filled microvias could obtain resistivities as low as 7.0 and 6.3 μΩ·cm under sintering temperatures of 220 °C and 260 °C, respectively. The property is comparable to that of the electroplated microvias, and much low than that of microvias filling with metallic particles in previous reports. The Cu@Ag NMPs and dense refill method in this study possess great potentials in future advanced electronic applications.

## Figures and Tables

**Figure 1 nanomaterials-12-01063-f001:**
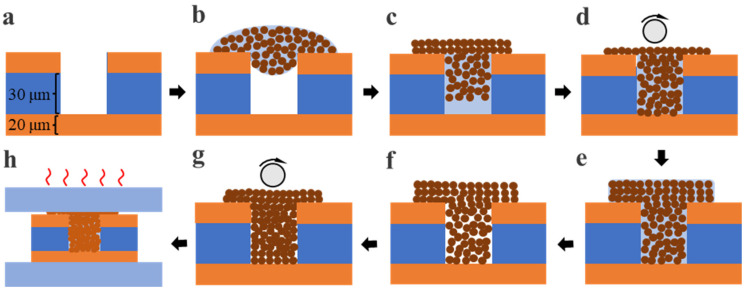
Procedures of the microvia filling process. (**a**) FCCL with a Blind via, (**b**) printing of Cu@Ag NMP paste, (**c**) evacuation, (**d**) roll pressing, (**e**) re-printing of Cu@Ag NMP paste, (**f**) drying, (**g**) roll pressing, (**h**) hot-pressing sintering.

**Figure 2 nanomaterials-12-01063-f002:**
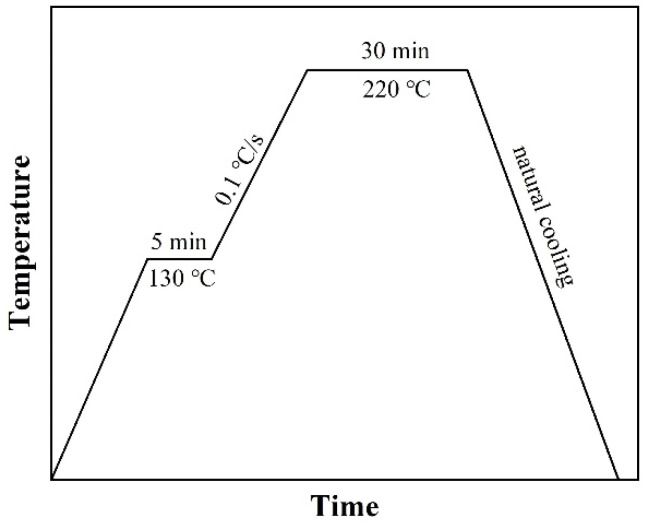
A typical temperature curve during the sintering process.

**Figure 3 nanomaterials-12-01063-f003:**
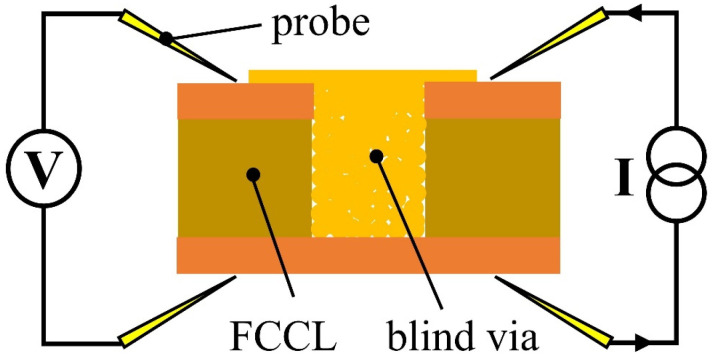
Schematic of the four-point probe measurement method.

**Figure 4 nanomaterials-12-01063-f004:**
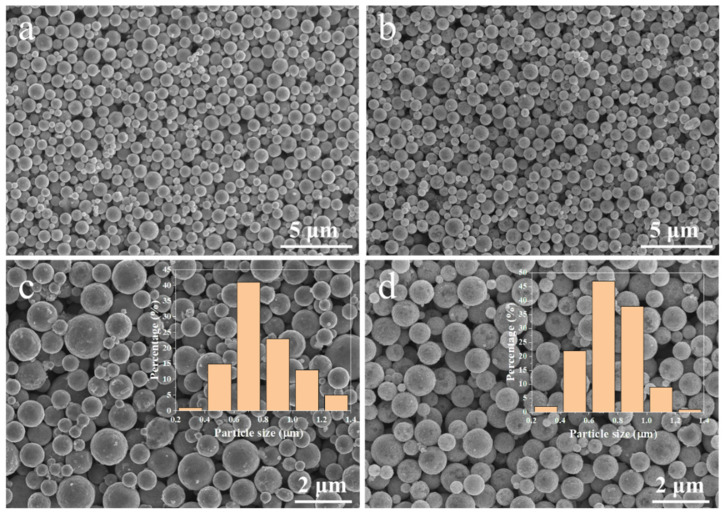
SEM images of (**a**,**c**) the as-synthesized Cu NMPs and (**b**,**d**) the as-synthesized Cu@Ag NMPs. The inserted figures show the particle size distribution.

**Figure 5 nanomaterials-12-01063-f005:**
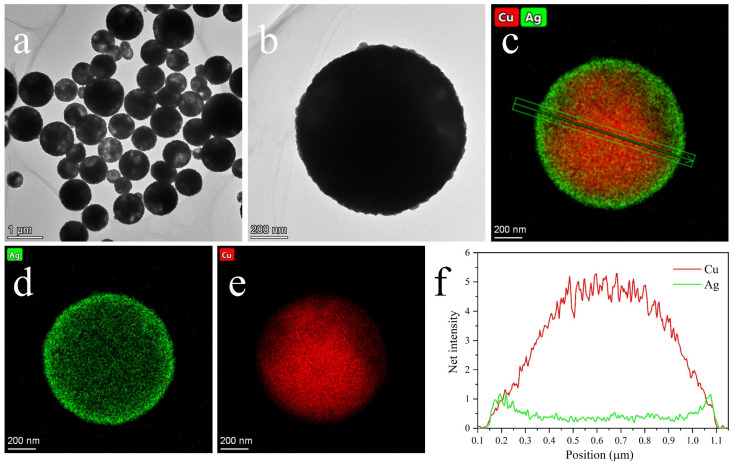
(**a**) TEM image of the as-synthesized Cu@Ag NMPs. (**b**) TEM image of an individual Cu@Ag particle. (**c**) EDS map of a Cu@Ag particle. (**d**) Ag element distribution. (**e**) Cu element distribution. (**f**) Cu and Ag element distribution curves along the diameter of Cu@Ag article.

**Figure 6 nanomaterials-12-01063-f006:**
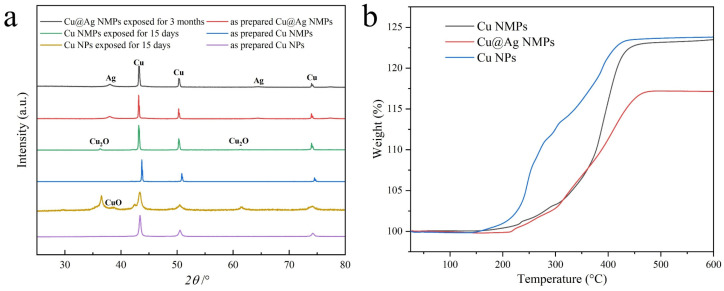
(**a**) XRD patterns of the as-synthesized Cu@Ag NMPs, the Cu NMPs and the Cu nanoparticles before and after exposure in air. (**b**) TG curves of the as-synthesized Cu@Ag NMPs, the Cu NMPs, and the Cu nanoparticles.

**Figure 7 nanomaterials-12-01063-f007:**
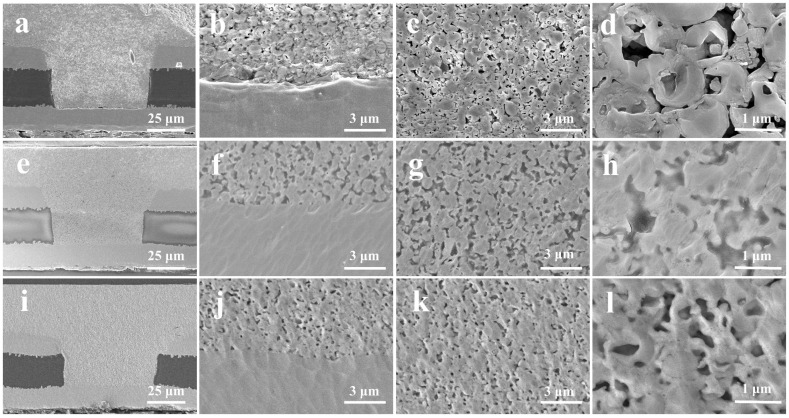
SEM images of the microvias filled with Cu@Ag NMPs at the sintering temperatures of 220 °C and sintering time of (**a**–**d**) 10 min, (**e**–**h**) 30 min, and (**i**–**l**) 50 min.

**Figure 8 nanomaterials-12-01063-f008:**
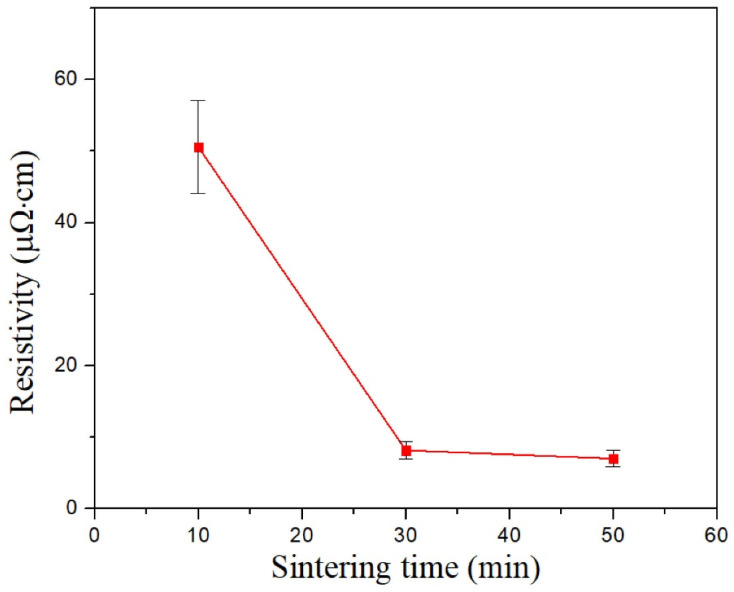
Electrical resistivities of the microvias filled with Cu@Ag NMPs at the sintering temperature of 220 °C and sintering time of 10 min, 30 min, and 50 min.

**Figure 9 nanomaterials-12-01063-f009:**
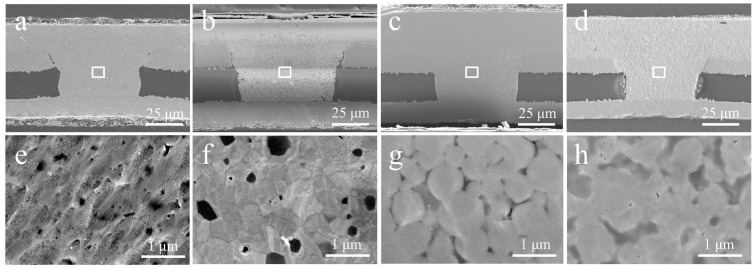
Cross-section images of the microvias filled with (**a**,**e**) Cu NPs, (**b**,**f**) Ag NPs, (**c**,**g**) Cu NMPs, (**d**,**h**) Cu@Ag NMPs at the sintering temperatures of 260 °C.

**Figure 10 nanomaterials-12-01063-f010:**
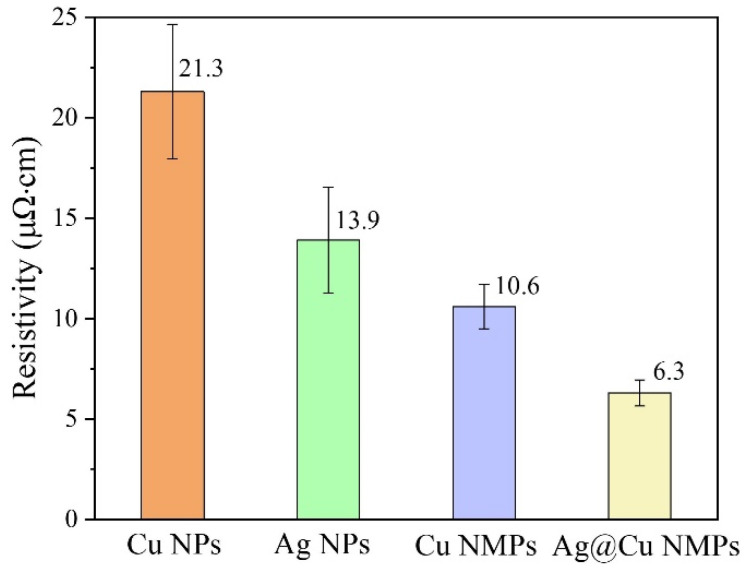
Electrical resistivities of the microvias filled with Cu NPs, Ag NPs, Cu NMPs and Cu@Ag NMPs at the sintering temperatures of 260 °C.

**Figure 11 nanomaterials-12-01063-f011:**
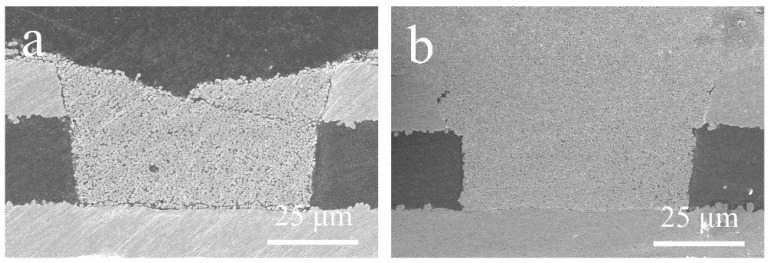
Cross-section images of the microvias filled with Cu@Ag NMPs at the sintering temperatures of 260 °C by (**a**) stencil printing method and (**b**) dense refill method.

## Data Availability

The datasets generated during and/or analyzed during the current study are available from the corresponding author on reasonable request.

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
