# Peer review of "A Green and Facile Microvia Filling Method via Printing and Sintering of Cu-Ag Core-Shell Nano-Microparticles"

_nanomaterials, 2022, doi:10.3390/nano12071063_

Round 1
Reviewer 1 Report
The paper with title “A green and facile microvia filling method via printing and sintering of Cu-Ag core-shell nano-microparticles” presents interesting results.
The paper is well written and well detailed.
The English is good.
However, there are some gaps and errors that should be solved.
- In the Introduction part, please underline the novelty and originality of this work;
- Page 3, line: 105: the authors said that “dried the sample in 80 °C gas flow for 3 min”. Please indicate the type of gas used;
- Page 3, lines 118-119: please replace “microscopy” with “microscope”;
- Page 4, lines 137-138: the authors wrote that the particles “are uniformly dispersed without obvious aggregation” but in order to affirm that, SEM images at on a larger area (smaller magnification) should be shown;
- Page 4, line 139-140: the sentence “TEM images of the as-synthesized Cu@Ag NMPs” has no verb; please correct it.
- Page 7, line 214: Before this line, the authors underlined the importance of the low temperature that they used (220 Celsius) and all the results and discussions referred at 220 Celsius. Even the conclusions referred at the sintering temperature of 220 °C. Starting with line 214, the authors presented data obtained at sintering temperature of 260 °C.
Is it a typing error?
If not, please explain why did you increase the temperature, and modify the whole manuscript for 260 Celsius.
- Page 7, line 232: “superb property” is too much…please mention this property.
Author Response
Dear reviewer,
Please see the attachment for the response letter with images.
This letter is in response to the reviewer’s report for the paper entitled “A green and facile microvia filling method via printing and sintering of Cu-Ag core-shell nano-microparticles” by Guannan Yang, et al. (nanomaterials-1613215). We are grateful for your careful reading of the manuscript and appreciate your comments. According to the comments, we have carefully revised our manuscript. Presented below are our point-by-point response to these comments.
- In the Introduction part, please underline the novelty and originality of this work.
Answer: We have added some content to the introduction part to explain the novelty and originality of this work
In this study, we try to improve the microvia filling performance of metallic particles from the perspective of material design and filling method. In terms of filling materials, Cu and Ag particles are widely used in the previous studies of conductive inks, interconnect materials and printed electronics [15,20,21]. However, Cu particles are easily oxidized in air, and Ag particles are prone to electronic migration, which severely limit their applications [22]. Here we propose a chemical reduction method to synthesize Cu@Ag NMPs as microvia fillers. The as-synthesized Cu@Ag NMPs can achieve the combined properties of oxidation and electronic migration resistance. The Cu@Ag NMPs are in a wide size range from 0.2~1.4 μm, which can reach higher packing density than uniform-sized particles. To further improve the filling compactness, the Cu@Ag NMPs are mixed with volatile, low-viscosity and decomposable ethylene glycol solvent to prepare a paste with a reactively high solid content (80 wt.%). Compared with other metal particle conductive inks with relatively low solid content (40~50 wt.% [23]), and metal particle conductive adhesives with undecomposable organic additives such as film-forming agents and cross-linking agents [24], the Cu@Ag particle paste is able to reach better sinterability and conductivity. In terms of filling method, previous studies have tried screen printing [25-27], squeegee printing [28], and inkjet printing [7]. However, the filling performance of these methods are unsatisfactory, as the evaporation of solvents will lead to shrinkage cavities in the structure. Here we develop a dense filling method for microvias. Through drying and refilling the microvias, a better filling compactness can be obtained. With the combined improvements in material and filling method, a promising microvia filling performance can be achieved.
- Page 3, line: 105: the authors said that “dried the sample in 80 °C gas flow for 3 min”. Please indicate the type of gas used;
Answer: the samples are dried in air gas flow.
- Page 3, lines 118-119: please replace “microscopy” with “microscope”;
Answer: we have corrected this mistake.
- Page 4, lines 137-138: the authors wrote that the particles “are uniformly dispersed without obvious aggregation” but in order to affirm that, SEM images at on a larger area (smaller magnification) should be shown;
Answer: We have provided the SEM images on a larger area of the two kinds of particles in our revised manuscript.
- Page 4, line 139-140: the sentence “TEM images of the as-synthesized Cu@Ag NMPs” has no verb; please correct it.
Answer: We have corrected the sentence.
Fig. 5 shows the TEM images of the as-synthesized Cu@Ag NMPs.
- Page 7, line 214: Before this line, the authors underlined the importance of the low temperature that they used (220 Celsius) and all the results and discussions referred at 220 Celsius. Even the conclusions referred at the sintering temperature of 220 °C. Starting with line 214, the authors presented data obtained at sintering temperature of 260 °C. Is it a typing error? If not, please explain why did you increase the temperature, and modify the whole manuscript for 260 Celsius.
Answer: This is not a typing error. We selected this sintering temperature because the filling performance of other materials (Cu NPs, Ag NPs, Cu NMPs) are poor at low sintering temperature (200~220 °C). In previous reports, Khorramdel et al. [3] tried to fill blind vias on silicon substrates with Ag NPs paste under the sintering temperature of 220 °C. But the resistivity of the filled vias are 2-4 orders of magnitude higher than bulk silver. Yang et al. [40] filled TSVs with silver NPs paste under the sintering temperature of 220 °C, and the resistivity of the filled vias is relatively high (400 μΩ·cm). With increasing sintering temperature, the sintering properties of metal particles can be significantly improved [41]. Quack et al. [42] filled TSVs with Ag NPs under the sintering temperature of 250 °C, and the resistivity is less than 200 μΩ·cm. In our previous work [41], we found that sintering strength of Cu MNPs increased by 100~230% when the sintering temperature increased from 220 °C to 260 °C. Under the sintering temperature of 220 °C, the MNPs are unable to form thick sintering necks with each other. Therefore, in this work, we chose the sintering temperature of 260 °C to better compare the filling performance of different materials and different filling methods. We have also explained this in our revised manuscript.
- Page 7, line 232: “superb property” is too much…please mention this property.
Answer: We have corrected the sentence.
The Cu@Ag NMPs combine the advantages of Ag shell and mixed particle sizes, thus exhibiting better conductivity among the four groups of particles.
We have carefully revised our manuscript according to your advice. All the changes are marked in the revised manuscript. We thank again for your careful reading and we really appreciate for your very helpful advice.

Reviewer 2 Report
The manuscript submitted by Guannan Yang and coworkers describe an eco-friendly and facile microvia filling method by using printing and sintering of Cu-Ag core-shell nano-microparticles (Cu@Ag NMPs). Though the authors reported the preparation and characterization in an order, manuscript can be accepted in the jurnal of Nanomaterials after overall English, and typo error check.
Author Response
Dear reviewer,
We are grateful for your review and positive comments on our manuscript. Thank you very much!
Sincerely,
Guannan Yang
School of Electromechanical Engineering,
Guangdong University of Technology,
Guangzhou 510006, China
Round 2
Reviewer 1 Report
The manuscript fulfils the reviewer’s comments.
I recommend to be accepted as it is.